# Dynamic modelling of cell cycle arrest through integrated single-cell and mathematical modelling approaches

Javiera Cortés-Ríos[1,2*], Maria Rodriguez-Fernandez[1], Peter Karl Sorger[3], Fabian Fröhlich[4*]

**1** Institute for Biological and Medical Engineering, Schools of Engineering, Medicine and Biological Sciences, Pontificia Universidad Católica de Chile, Santiago, Chile, **2** Department of Pharmaceutical Sciences, School of Pharmacy and Pharmaceutical Sciences, University at Buffalo, State University of New York, Buffalo, New York, United States of America, **3** Laboratory of Systems Pharmacology and Department of Systems Biology, Harvard Medical School, Boston, Massachusetts, United States of America, **4** Dynamics of Living Systems Laboratory, The Francis Crick Institute, London, United Kingdom

\* fabian.frohlich@crick.ac.uk (FF); jacortes4@uc.cl (JC-R)

## Abstract

Highly multiplexed imaging assays allow simultaneous quantification of multiple protein and phosphorylation markers, providing a static snapshots of cell types and states. Pseudo-time techniques can transform these static snapshots of unsynchronized cells into dynamic trajectories, enabling the study of dynamic processes such as development trajectories and the cell cycle. Such ordering also enables training of mathematical models on these data, but technical challenges have hitherto made it difficult to integrate multiple experimental conditions, limiting the predictive power and insights these models can generate. In this work, we propose data processing and model training approaches for integrating multiplexed, multi-condition immunofluorescence data with mathematical modelling. We devise training strategies for mathematical models that are applicable to datasets where cells exhibit oscillatory as well as arrested dynamics and use them to train a cell cycle model on a dataset of MCF-10A mammary epithelial cells exposed to cell-cycle arresting small molecules. We validate the model by investigating predicted growth factor sensitivities and responses to inhibitors of cells at different initial conditions. We anticipate that our framework will generalise to other highly multiplexed measurement techniques such as mass-cytometry, rendering larger bodies of data accessible to dynamic modelling and paving the way to deeper biological insights.

## Author summary

Advanced imaging techniques allow us to see detailed pictures of different proteins and cell changes. By using computational algorithms, we turn these static

**Data availability statement:** MATLAB scripts for model calibration, simulation, and data analysis can be reached at https://www.synapse.org/#!Synapse:syn55267562/files/.

**Funding:** This work was supported by NCI grant U01-CA284207 and the Ludwig Cancer Center at Harvard to P.K.S. Additionally, it was supported by the National Agency for Research and Development (ANID) under Grants Fondecyt 1230844, ACT210083 and Millennium Science Initiative Program - ICN2021_004 to M.R.F., and J.C.R. was sponsored by ANID / Scholarship Program / DOCTORADO / 2019 — 21191120. F.F. is supported by the Francis Crick Institute, which receives its core funding from Cancer Research UK (CC2242), the UK Medical Research Council (CC2242), and the Wellcome Trust (CC2242). The funders had no role in study design, data collection and analysis, decision to publish, or preparation of the manuscript.

pictures into dynamic sequences to understand processes like the cell cycle better. However, combining data from different experiments is difficult and limits how well our models can predict outcomes. This study introduces new ways to process data and train models to handle complex data from various conditions. The approach is tested by using data from untreated and treated cells to create a model of the cell cycle. This model was then checked for accuracy by seeing how well it could predict how cells respond to growth factors and drugs from different starting points. In the future, this method could be used with other data types, allowing for more detailed and accurate models of cellular behavior.

## Introduction

Using mathematical models of cellular processes to understand, predict, and optimize biological processes and pharmacological responses has become an important tool for generating and testing hypotheses and developing new therapies [1–3]. For these models to develop their full potential as descriptions of the biological mechanisms underlying such phenotypes, these models must be trained on experimental data that capture dynamic responses to perturbation, such as stimulation with growth factors or small molecules. Traditionally, such data has been generated using relatively low-throughput techniques such as Western Blotting - which is limited to a few timepoints or perturbations - or live-cell imaging, which measures only a small number of cellular features [4–7]. In contrast, highly multiplexed assays involving immune reagents, such as multiplexed immunofluorescence [8–10] or mass cytometry [11] make it possible to measure multiple proteins and their modifications in single cells across a range of conditions or time points. Such methods are suitable for assaying "many conditions" and serve as a natural complement to approaches such as single-cell RNAseq [12,13], that are potentially transcriptome or proteome scale, but limited by cost and complexity to relatively few experimental conditions. Time resolved measurements are particularly important when developing mechanistic and semi-mechanistic mathematical models [14], but rarely available for high-throughput assays.

The lack of time-course data in many single-cell datasets can be overcome using pseudo-time ordering techniques, which reconstruct dynamic trajectories for individual cells from static snapshots. These techniques exploit the fact that unsynchronised cultures contain cells at many different points in a process. The underlying temporal trajectories can be reconstructed in this case using methods such as diffusion maps [15], Potential of Heat-diffusion for Affinity-based Trajectory Embedding (PHATE) embeddings [16] or classical multidimensional scaling (CMD scaling) [9,10,15,17,18], with some temporal ordering, which can be based on Markov Theory [15,17,19,20], optimal transport [21], graph abstraction [22] and other methods. Such pseudo-time ordering techniques have enabled the training of mathematical models defined by ordinary differential equations (ODEs) [23,24] using single cell data collected at a single point in time.

Most applications of pseudo-time approaches have focused on trajectory inference in developmental systems [25,26] and the cell cycle [27–30]. An ordered cell cycle is not only essential for all eukaryotic cells, but also a well understood system involving multiple sequential states (G0, G1, S, G2 and M phases) and well-defined protein markers, such as cyclins, which both report on cell state and control cell cycle progression. Unlike developmental systems, the cell cycle is generally assumed not to involve branching trajectories, making it easier to model. Studying the cell cycle also has significant potential for impacting human health, as many cancer drugs induce cell cycle arrest or cell death. The eucaryotic cell cycle has been the subject of extensive mathematical modelling over the years. These models include descriptions of the cell cycle machinery ranging at different scales from small oscillatory circuits [31,32] to complex descriptions involving multiple regulatory pathways [33–35], as well as simpler, phenomenological models [36–39]. A recent study by Lang et al. [23] is the first to train mathematical models of the cell cycle on trajectories reconstructed from highly multiplexed immunofluorescence (IF) measurements made under a single condition. Highly multiplexed IF is a means of collected data on 10–40 protein levels, localizations, and modifications at a single cell level from cell grown in culture [40] or from tissues [8]. Lang et al. used data on 16 molecular markers measured in 300 single cells and found that simulations were overly constrained during model training, limiting the predictive power of the model and potentially reducing the biological insight.

These limitations can potentially be addressed by training a model on data from multiple experimental conditions that correspond to defined perturbations with small molecules or genetic manipulation. However, in the context of the cell cycle, the combination of stable oscillations in unperturbed conditions and cell cycle arrest under perturbed conditions complicates data integration. Such a combination of dynamic regimes makes it challenging to generate consistent pseudo-time orderings across conditions, since it is unclear if joint embeddings meet the ergodic properties assumed by most ordering methods. Perturbations that block cell cycle progression also introduce difficulties in training of mathematical models because they impose both oscillatory and steady-state constraints. These constraints are already difficult to manage individually [41,42] and there are no established methods for training models featuring both types of constraints.

In this work, we extend a cc-CMD pseudo-time ordering approach [9] to multi-condition data, thereby enabling pseudo-time ordering of perturbational data. Additionally, we develop a novel training technique for mathematical models that can incorporate oscillatory and steady-state constraints. We apply the extended cc-CMD approach along with this training method for mathematical models to multiplexed single-cell immunofluorescence data from MCF-10A cells [9], enabling the training of a mechanistic model of the cell cycle under both untreated and drug-induced arrested conditions by palbociclib and nocodazole. We demonstrate that only the trained model on both conditions accurately replicates oscillatory and arrested cell cycle dynamics in training data and predicts cell arrest states for cells at different starting points of the cell cycle as well as G1 arrest after growth factor depletion.

## Results

### Processing of multiplexed immunofluorescence data allows to reconstruct the dynamics of the main cell cycle proteins

To generate pseudo-time-ordered cell cycle progression trajectories controlled by core cell cycle regulatory proteins, we applied the CMD dimensionality reduction method to a published dataset that captured cell cycle marker dynamics in unsynchronized, unperturbed MCF-10A cells using multiplexed immunofluorescence imaging [9]. Following the approach in the original manuscript, the pre-processed intensities of each marker and cell were log-scaled and normalized.

Since experimental noise in single cell data and artifacts specific to measurement of some marker proteins can negatively impact pseudo-time ordering, some studies, including the original Gaglia et al. [9] manuscript, focus on an empirically chosen subset of markers to generate the CMD embedding. To make this selection process more rigorous and systematically assess the impact of individual markers on the ordering, we performed a paired t-test across multiple CMD embeddings after removal of individual markers. To ensure the robustness of the method but maintain computational

tractability, we down-sampled the data by focusing on a randomly selected subset of 12,000 MCF-10A cells. Statistical tests showed no significant differences (p-values > 0.05, using the Benjamini & Hochberg (1995) procedure for controlling the false discovery rate for multiple-testing [43], S1 Fig), indicating that the CMD embedding is robust to the removal of individual markers. The same analysis was performed on published biopsy datasets for ER +, triple-negative (TNBC), and HER2 + breast cancer samples [9] which also showed no significant changes upon removal of a single marker (S2 Fig). We therefore used all available cell-cycle markers to perform the CMD; these corresponded to cyclin D, cyclin E (excluded for ER+ and HER2 + samples due to missing data), cyclin A, cyclin B, pRB, CDT1, geminin (Gem), p21, p27, Ki67, PCNA, and MCM2.

Applying CMD scaling to 12,000 MCF-10A cells yielded a two-dimensional embedding (Fig 1A) in which the coordinates of individual cells (blue dots) collectively formed a circle, reminiscent of the cyclic progression of cells through different stages of the cell cycle. To extract dynamic marker trajectories (Fig 1B) from this embedding, we followed the original approach and computed the moving median (solid red lines) and moving median absolute deviation (dashed blue lines) across all individual cells (blue dots) after inferring angular position (CMD angle) from two-dimensional coordinates. We observed a non-uniform CMD angle distribution (Fig 1C, **top**) with two major modes. We hypothesized that these modes correspond to (aggregated) cell cycle phases. Using relative cyclin levels, we manually annotated the minima between these modes as M/G1 and S/G2 phase transitions of the cell cycle based on a review by Hochegger and colleagues [44]. Cell cycle time (Fig 1C, **bottom**) was reconstructed by assigning equidistant time points to the sorted angles within the interval from 0 to $2\pi$, yielding a uniform distribution of cell cycle times. This approach assumes that the sample of cells analysed contains cells in different cell cycle phases, with population densities proportional to the time spent in each phase (mean sojourn time), a common assumption in single-cell data analysis (this is effectively an assumption that the data are ergodic [45]). We note that this assumption has been challenged in the context of cells not grown to full confluency [23,27], but found that time inference methods that do not rely on ergodicity yield comparable reconstructed marker trajectories, with negligible influence on downstream analyses (S3 Fig and S1 Table).

To validate our trajectory inference method, we investigated reconstructed trajectories for cyclin A/B/D/E markers (Fig 1D). We observed that the obtained cyclin dynamics closely match the expected cell cycle progression, with Cyclin D (blue curve) high in G1 to G2 phases, cyclin E (light blue curve) high in late G1 and early S phases, cyclin A (light yellow curve) high in S and G2 phases, and cyclin B (light orange curve) high in late G2 and M phases [44,46]. This approach correctly ordered cell cycle phases as being G1 followed by S followed by G2 followed by M. Furthermore, the local minima in the angle distribution (Fig 1C) were observed to roughly coincide with the M/G1 phase transition (point of minimum cyclin expression) and the S/G2 phase transition (just prior to the peak of Cyclin A) [44]. Subsequently, cell cycle time inference suggested that cells spend approximately two thirds of the cell cycle time in G1 + S phases, and one third in G2 + M phases, which is consistent with prior reports with for MCF-10A cells [47,48]. In contrast, trajectory construction for patient-derived data, characterized by higher noise, fewer single cells (<7,000 in ER+ and HER2 + samples) and lower coverage of cell-cycle markers (missing cyclin E in ER+ and HER2 + samples), was less reliable and showed poorer agreement with previous reports (S1 Text and S4 Fig), and was therefore excluded from further analysis in this study. Overall, this confirms that, given sufficient data quality and quantity, CMD-based pseudo-time ordering of cell-cycle data is robust and generates biologically plausible trajectories.

## Drug-induced cell arrest states can be determined through CMD embedding

While it is natural to expect proliferating cells to adopt a circular pattern in embedding space, arrested cells usually adopt a unimodal distribution [9], complicating the identification of cell cycle phases that cells are arrested in. To better identify arrest phases, we constructed a joint embedding of treated and untreated cells and compared to individual embeddings of treated and untreated cells (S5 Fig). Even though we observed some changes to the embeddings of treated cells, potentially indicating better identification of arrest phases, we also observed substantial shifts in the cell cycle distributions of

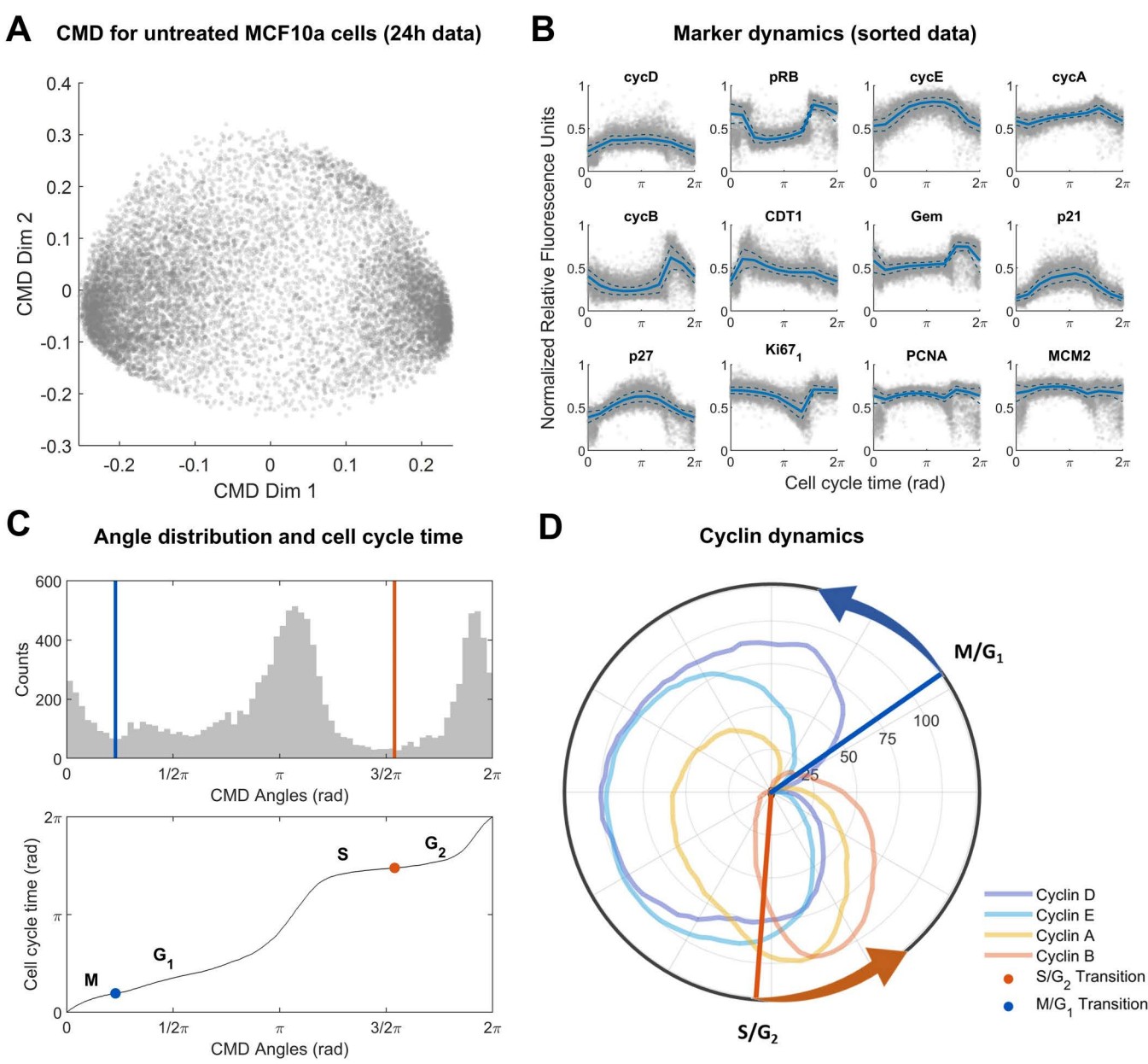

**Fig 1. Data analysis for untreated data.** (A) CMD embedding for untreated dataset of MCF-10A cells. Every dot corresponds to a single cell. (B) Reconstructed marker dynamics for cell cycle related genes. Solid lines indicate the moving median. Dashed lines indicate the median absolute deviation added or subtracted from the moving median. Individual cells are indicated as grey dots. (C) Cell cycle phase and pseudo-time reconstruction. Top panel: Histogram of CMD angles distribution (cell cycle phase). Local minima between major modes are indicated by coloured vertical lines. Bottom panel: Mapping between CMD angles and cell cycle time (pseudo-time). Inflection points of the mapping corresponding to local minima in the angle distribution (top panel) are indicated by coloured dots. The local minima in the top panel were annotated as M/G1 and S/G2 cell cycle transitions based on Hochegger *et al.* [44]. (D) Polar coordinate plot of normalized cyclin dynamics. Radial coordinate corresponds to cell cycle time from (C). Polar coordinate corresponds to moving median of marker levels from (B) min-max normalized to the interval [0,100]. Coloured vertical lines correspond to the local minima from (C).

untreated cell (S6 Fig), suggesting that this integration strategy leads to biases in the reconstruction of cell cycle dynamics under unperturbed conditions. We therefore devised a new data integration strategy to generate embeddings and cell cycle phases that are consistent across multiple experimental conditions.

To preserve the embedding and ordering of untreated cells, we computed embeddings and corresponding CMD angles and cell cycle times for all untreated data plus small subsets (5%) of treated data. This process was repeated for each subset derived from a disjoint partition of the treated data (Fig 2A), yielding joint CMD embedding results (Fig 2B) for untreated (grey), nocodazole treated (light green, left panel) and palbociclib treated (purple, right panel) cells. The CMD angle distributions for nocodazole (Fig 2C) and palbociclib (Fig 2D) were unimodal, indicating cell cycle arrest in most cells. Evaluation of cell cycle times (Fig 2E) of corresponding modes (light green/purple dots, Fig 2C **and** 2D) suggested that nocodazole (light green line) and palbociclib (purple line) arrests cells in early G1 phase. This observation aligns with prior reports about the canonical mechanism of action of palbociclib, and non-canonical slippage and subsequent G1 arrest at low concentrations of nocodazole [49–51]. Therefore, we conclude that the proposed approach enables estimation of biologically plausible pseudo-time orderings across perturbed and unperturbed experimental conditions.

## The cell cycle model captures the proliferative dynamics and drug-induced cell arrest states for nocodazole and palbociclib

To integrate cell proliferation dynamics with drug-induced cell arrest results obtained through CMD scaling, we developed a dynamic cell cycle model consisting of 6-ODEs derived from the complex and skeleton models of Gérard & Goldbeter [33,37]. We extended the original skeleton model by incorporating the p21/p27 regulation of the CDK-cyclins activity [33], the indirect regulation of cyclin B on the degradation of p21 and its indirect effect on phosphatase mediated pRB dephosphorylation [52–55], and the effects of palbociclib and nocodazole on cell cycle components based on their respective mechanisms of action [51,56] (Fig 3, interactions not included in Gérard & Goldbeter models are highlighted in bold). Notably, this model enables intrinsic oscillation without the inclusion of growth factor pulses or other external inputs. We found that the incorporation of the inhibitory regulation of p21/p27 on CDK-cyclins was essential for representing CDK-cyclin activity across various components of the differential equations, while cyclin B regulation on p21 degradation [54,55], a factor not addressed in the Gérard & Goldbeter models, proved crucial in capturing the effect of nocodazole (see Methods for a more comprehensive description of the model and its equations). Structural identifiability assessment using a sensitivity matrix analysis approach [57] (*StrucID* method in Data2Dynamic environment of MATLAB) revealed that our extended model is structurally identifiable. Thus, the model incorporates essential regulatory mechanisms governing cell cycle progression and drug effects while maintaining an identifiable structure.

Data from both untreated and treated conditions were used for parameter estimation. The proposed model training approach builds on maximum likelihood estimation, with specific conditions for fitting oscillatory systems under drug-induced arrest. Specifically, to encourage the model to exhibit stable oscillations under treated conditions, we concatenated four copies of the dynamic trajectories of all markers. To encourage model simulation to converge to a steady state under treated conditions, we concatenated 10 temporally equally spaced copies of the moving average marker concentrations at the cell cycle time of the corresponding arrest between the end of the first and the fourth cell cycle. In both cases, the copies of the data can be interpreted as regularization terms (Equations 7–9) that are added to the standard log-likelihood function to implement soft constraints for steady-state and oscillatory behavior. In contrast to strict enforcement via steady-state or periodic boundary constraints, this approach is trivial to implement. Moreover, it allows evaluation of the loss function even when the model does not exhibit stable oscillations or converge to a steady state, which enables wider exploration of parameter space during model training.

To recapitulate the asynchronicity of cell cycle phases at treatment time, we used the two most prominent modes of the cell cycle time distribution for untreated data to generate two sets of initial conditions. For variables that were directly measured ($p21$ and cyclins D, E, A and B) we used average marker concentrations at the mode, while $E2F_a$ initial conditions

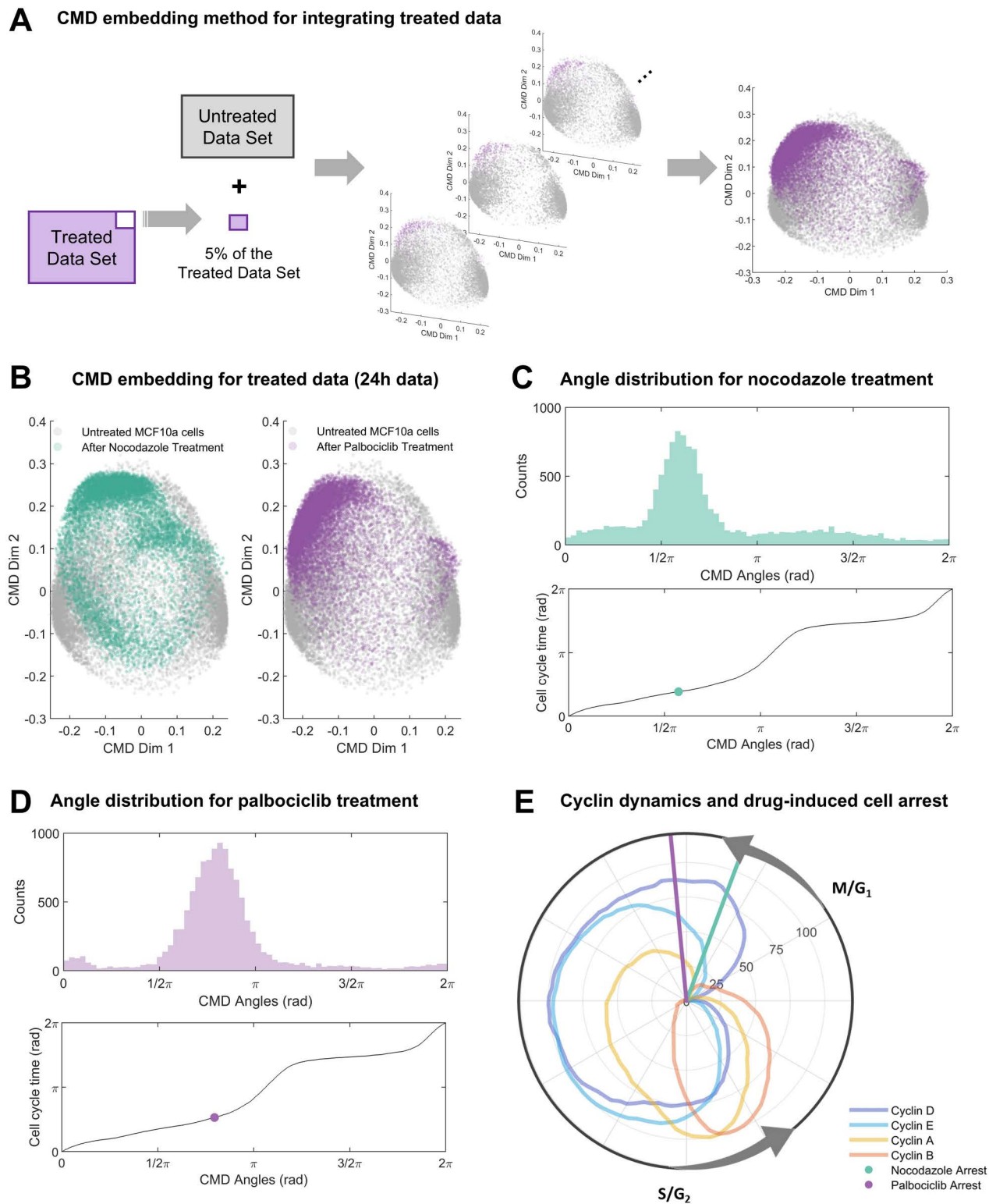

**Fig 2. Data analysis for treated data.** (A) CMD embedding method to estimate the cell arrest angle of treated MCF-10A cells. Untreated data plus subsets of 5% of the treated data were used to perform CMD embedding until obtaining the angles for all treated cells. Every dot corresponds to a single cell. (B) Joint CMD embedding results for MCF-10A cells treated with nocodazole on the left and palbociclib on the right. (C) Cell cycle phase and

pseudo-time reconstruction for nocodazole treated cells. Top panel: Histogram of CMD angles distribution (cell cycle phase). Bottom panel: Mapping between CMD angles and cell cycle time (pseudo-time). The green dot corresponds to the mode of the cell arrest angle. (D) Cell cycle phase and pseudo-time reconstruction for palbociclib treated cells. Top panel: Histogram of CMD angles distribution (cell cycle phase). Bottom panel: Mapping between CMD angles and cell cycle time (pseudo-time). The purple dot corresponds to the mode of the cell arrest angle. (E) Polar coordinate plot of normalized cyclin dynamics (untreated cells) and cell arrest times for nocodazole and palbociclib. Radial coordinate corresponds to cell cycle time from untreated cell dynamics. Polar coordinate corresponds to moving median of marker levels min-max normalized to the interval [0,100]. Transitions of the cell cycle phases G1/M and S/G2 were annotated based on [44] and results obtained previously (Fig 1).

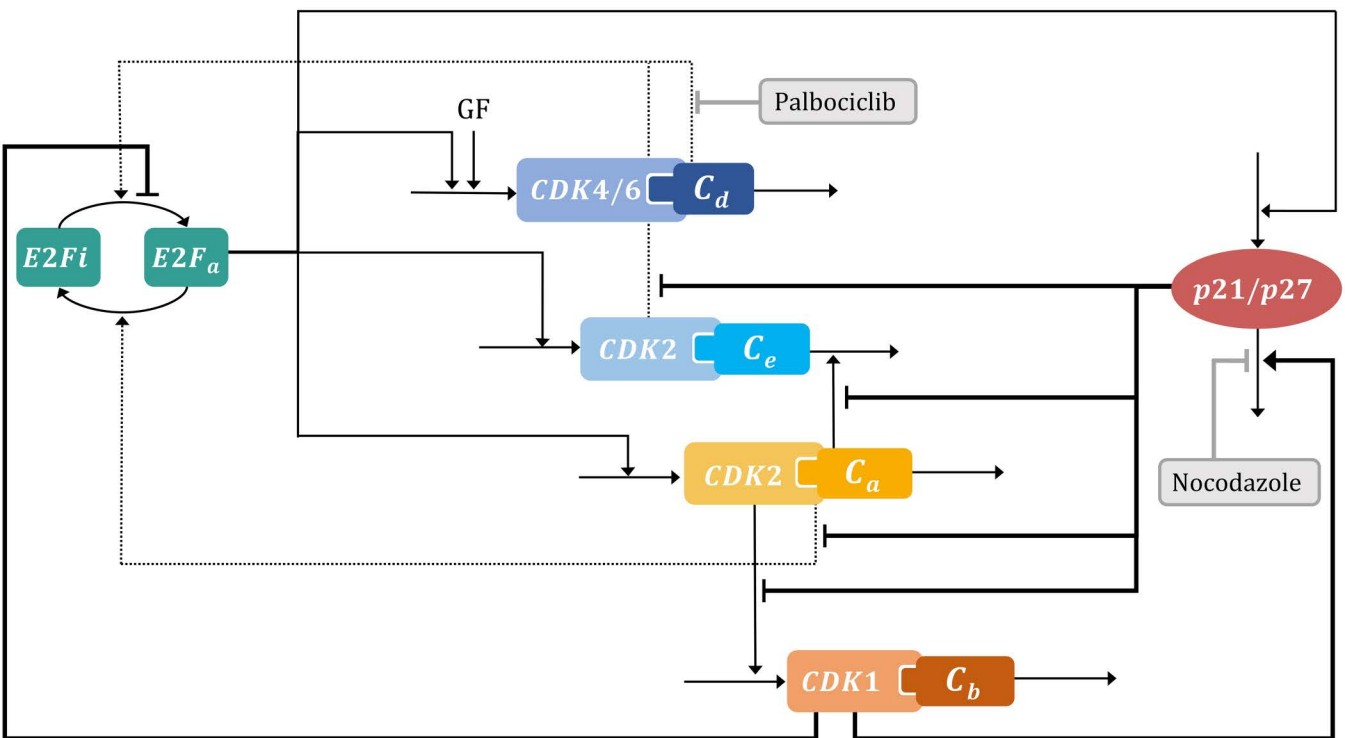

**Fig 3. Cell cycle model scheme.** The cell cycle model based on the skeleton model developed by Gérard and Goldbeter [37] includes the interaction between E2F transcription factor, growth factors, CDK-cyclins and p21/p27 proteins. Each black line or dashed arrow represents the direct or indirect positive effect on the other and the blunt head arrows indicate direct or indirect inhibitory effects. Inhibitory effects of palbociclib and nocodazole are incorporated in grey blunt head arrows. p21/p27 is included as one entity. New interactions: CDK2 – cyclin E/A inhibitory effect of p21/p27, indirect inhibitory effect of CDK1 – cyclin B on E2F activation and p21/p27 degradation are highlighted in bold. $E2F_i$: E2F inactive, $E2F_a$: E2F active, $GF$: growth factor, $C_d$: cyclin D, $C_e$: cyclin E, $C_a$: cyclin A, $C_b$: cyclin B. $CDK4/6$: cyclin-dependent kinase-4/6, $CDK2$: cyclin-dependent kinase-2, $CDK1$: cyclin-dependent kinase-1, $p21/p27$: cyclin-dependent kinase inhibitors.

were estimated (all estimated parameter values are shown in S1 Table). For dynamic trajectories in untreated conditions (Fig 4), the trained model (black lines) replicated oscillations observed in the experimental data (blue dots). Specifically, we observed the strongest agreement between model simulations and experimental data for p21, cyclin D, and cyclin E. For these markers, the model simulations consistently fall within the range of the median absolute deviations across all the cell cycle time. In contrast, the simulations for cyclin B and cyclin A were slightly outside the median absolute deviation range for a few datapoints. For treated conditions (Fig 5), we observed similarly good fits with all model simulations within the range of median absolute deviation for all datapoints (orange error bars) and markers. Assessing the impact of initial conditions, we found that the model consistently predicted convergence to a single steady state, both from the primary mode of CMD angles (black line, used for model training) and other points in the cell cycle (grey lines, not used for model

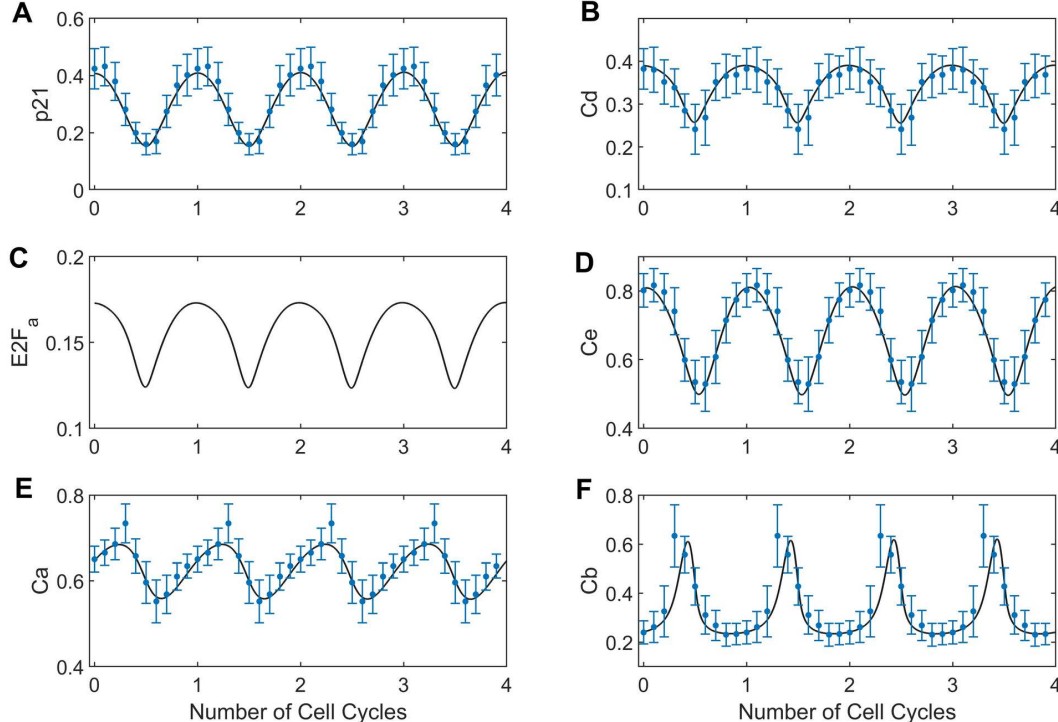

**Fig 4. Model fitting to cell cycle marker dynamics obtained from MCF-10A cells processed data.** Four cell cycles of experimental data obtained from processed data of untreated MCF-10A cells is shown in blue dots (moving median) with their respective median absolute deviations for each marker; (A) cyclin-dependent kinase inhibitor, $p21$; (B) cyclin D, $C_d$; (C) E2F active, $E2F_a$; (D) cyclin E, $C_e$; (E) cyclin A, $C_a$; (F) cyclin B, $C_b$. The black line corresponds to the model simulation for each marker.

training). This aligns with the experimental observations for palbociclib and nocodazole, where we observed that unsynchronized cells arrest at similar cell cycle phases after 24 hours of adding either drug (Fig 2B).

To assess parameter uncertainties, we used the profile likelihood approach [58]. For all model parameters except $V_{sd}$ and $V_{sb}$, which were partially identifiable, parameters were found to be practically identifiable as shown in S7 Fig. Conversely, when the model was fitted using only data from cells not treated with drugs, the parameters $V_{sd}$, $V_{sb}$, $kdd$, $kdd_1$, $kde$ and $kpi$, were practically non-identifiable (S8 Fig). Overall, these results show that we could construct a structurally identifiable mechanistic model, and we reduced the uncertainty in estimated parameters by incorporating experimental observations from treated and untreated conditions.

## Mathematical model predicts a cell cycle arrest in G1 phase for low GF values

To validate the model, we evaluated its ability to predict cell cycle arrest under starvation conditions—an experimental setting that was not included in the model training process. Using the proposed approach with the model fitted using untreated and treated datasets, we found that with growth factor levels at ~ 80% of training conditions, the model consistently stopped exhibiting stable oscillations and converged to an arrest condition. We established that the arrest condition corresponded to an arrest in early G1 (Fig 6A and black line in Fig 6B), consistent with prior experimental evidence. Furthermore, model simulations for 100 cell cycles (Fig 6C) at 60% (panel A), 80% (panel B), 95% (panel C) and 100% (panel D) of growth factor levels revealed that oscillations at high levels are stable beyond the four cell cycles included in the training data and that decreases in growth factor levels initially dampen the amplitude of cyclin oscillations and then

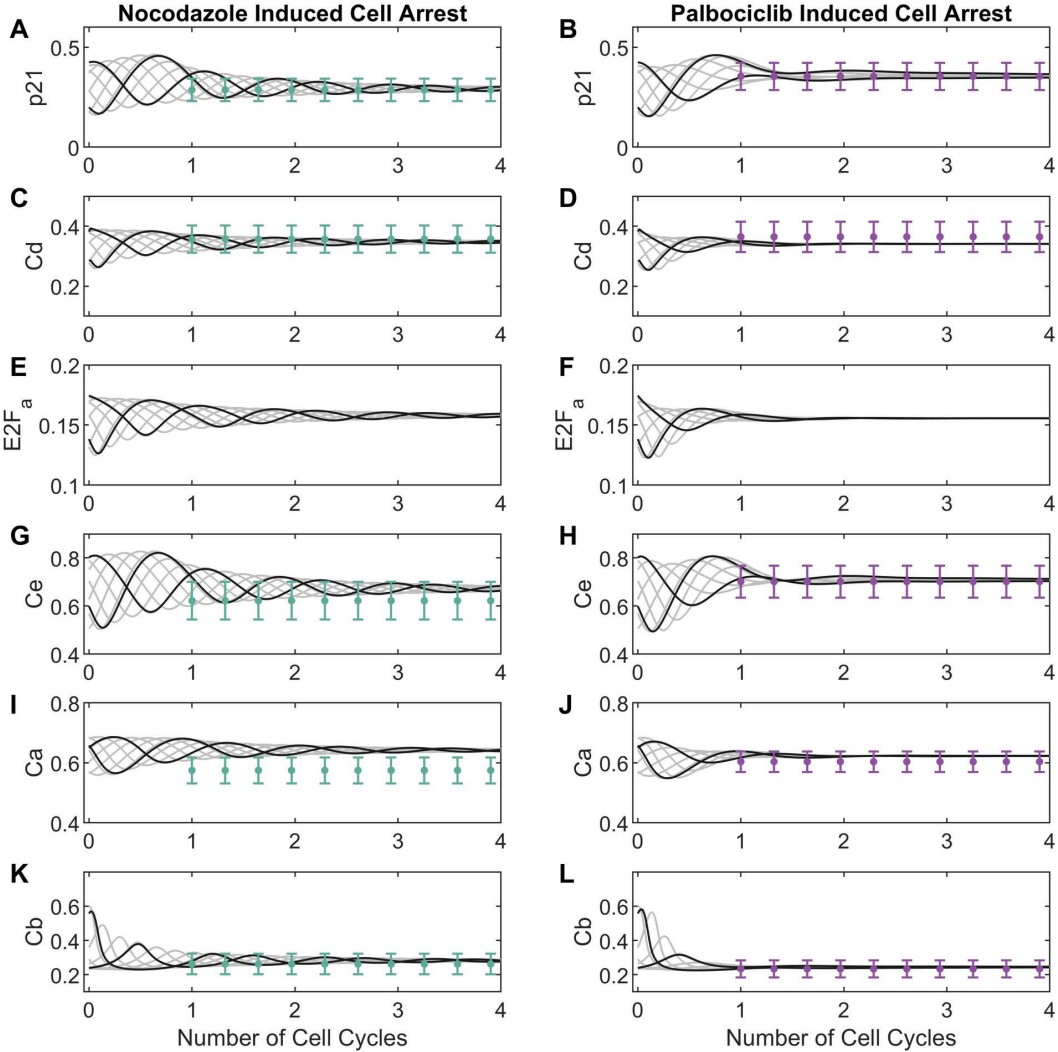

**Fig 5. Model simulations and adjustment of final arrest state after 24 hours of nocodazole (left) or palbociclib (right) treatment for MCF-10A cells.** The cell cycle arrest states obtained after 24 hours of treatment (approximately the time of one cell cycle) with nocodazole (on the left) or palbociclib (on the right) is shown in green/purple dots with their respective median absolute deviations across three cell cycles. Cell arrest data for two initial conditions, corresponding to the mode of the main subpopulations of Fig 1C (shown as black lines), were used to fit the model to the following markers: (A – B) cyclin-dependent kinase inhibitor, $p21$; (C – D) cyclin D, $C_d$; (E – F) E2F active, $E2F_a$; (G – H) cyclin E, $C_e$; (I – J) cyclin A, $C_a$; and (K – L) cyclin B, $C_b$. Model simulations (non-fitted to data) using multiple starting points as initial conditions are shown in grey lines for each marker.

accelerate convergence to arrest conditions. Unexpectedly, we found that lower growth factor levels shortened, rather than prolonged [59,60], the cell cycle length (S9 Fig). This is unlikely to be correct and we conclude that other potentially nutrient-related, regulatory mechanisms and corresponding training data will be necessary to further improve the biological plausibility of the model. Despite these limitations, we find that our model can predict biologically plausible responses for conditions not considered during model training. In contrast, when the model was fit only to the untreated dataset, simulations exhibited minimal sensitivity to GF reduction with oscillations even at a GF level of 1% of training conditions. Moreover, this "no-perturbation" model predicted a cell arrest state closer to the S phase for GF = 0.0001%, which is inconsistent with reported experimental results (S10 Fig). Thus, our results demonstrate that training on data from multiple

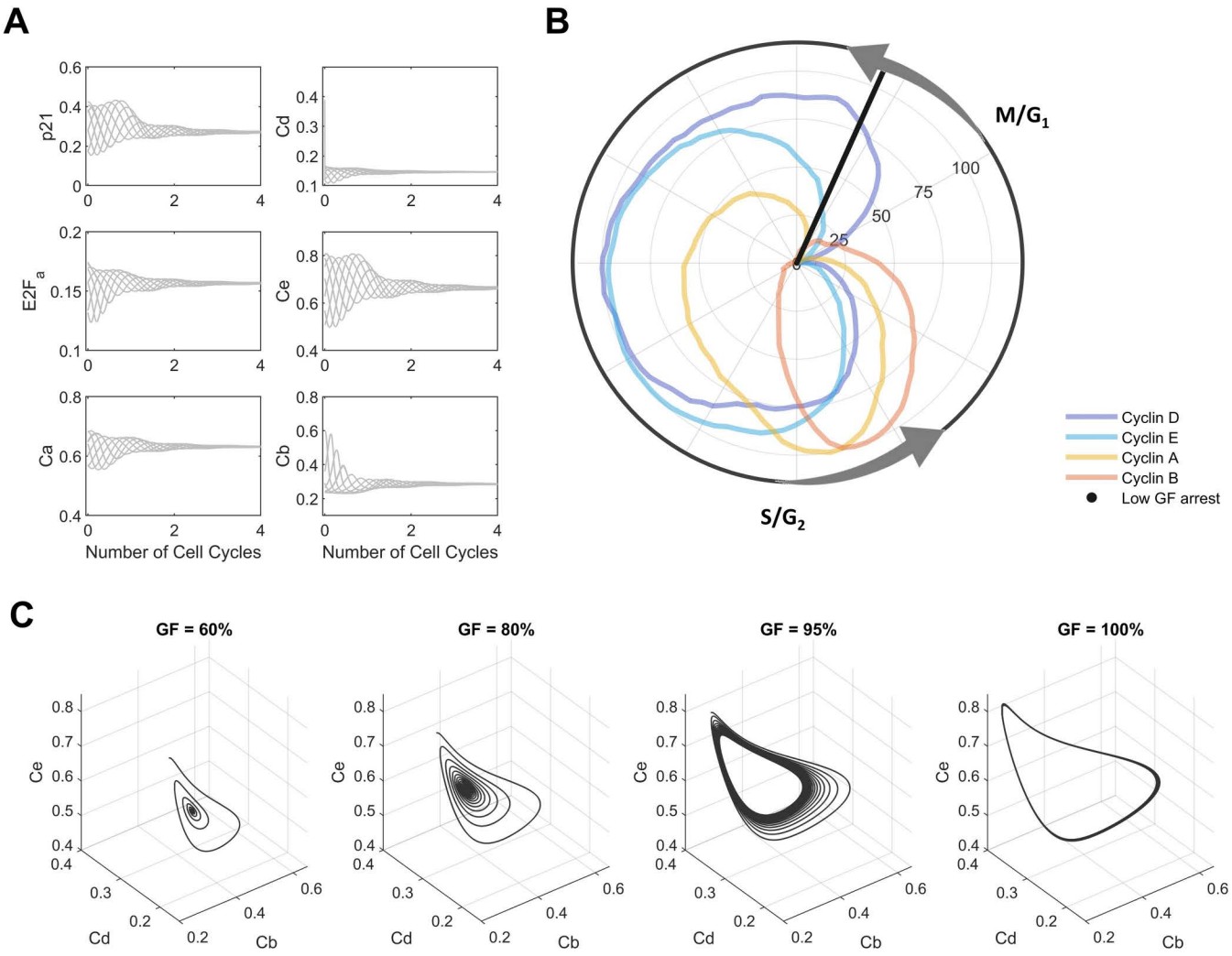

**Fig 6. Model predictions for low growth factor levels.** (A) Predicted dynamic of the state variables $p21$, $Cd$, $E2F_a$, $Ce$, $Ca$ and $Cb$ for different starting points of the cell cycle (initial conditions) lead to the same cell arrest state for low growth factor levels (60% of GF). (B) Polar coordinate plot of normalized cyclin dynamics (untreated cells) and arrest time predicted for low growth factor levels (60% of GF) at early G1 phase. Polar coordinate corresponds to moving median of marker levels min-max normalized to the interval [0,100]. Transitions of the cell cycle phases M/G1 and S/G2 were annotated based on results obtained previously (Fig 1). (C) Simulations for for $Ce$, $Cd$ and $Cb$ variables at different growth factor (GF) levels (60, 80, 95 and 100%) reveal a limit cycle for high GF values (≥ 80%) and an early G1 arrest at low GF values.

experimental conditions, particularly when these conditions provide informative variable dynamics, make it possible to significantly reduce parameter uncertainty and build a biologically plausible model that performs correctly on a least at subset of conditions outside the training set.

## Discussion

Highly multiplexed single-cell assays offer a new way to study cellular processes in cultured cells and tissue but usually require pseudo-time ordering to make data amenable to mathematical modelling. In this manuscript, we extend cc-CMD [9], an existing method for pseudo-time ordering, to enable integration of data from multiple experimental conditions; we also develop training methods for mathematical models that can handle mixtures of steady-state and oscillatory

constraints. We use the resulting framework to extend a cell cycle model and train it on data from MCF-10A cells under treated and untreated conditions and demonstrate that it reduces parameter uncertainty and is necessary to obtain a biologically plausible model.

The cell cycle model we developed was inspired by the skeleton model of Gerard & Goldbeter [33,37], but it includes additional mechanisms, such as the effect of p21 on CDK-cyclin activity, as well as the cyclin B effect on p21 degradation and its indirect effect on phosphatase-mediated pRB dephosphorylation [52–55]. Incorporating these mechanisms was crucial for recapitulating cell cycle dynamics and drug-induced cell arrest; however, other mechanisms might have explained the data equally well. A more rigorous understanding of which molecular mechanism contribute to cell cycle progression would likely require a richer dataset. Specifically, bespoke marker panels that capture the protein levels of cyclin dependent kinases as well as the activities of corresponding complexes would be beneficial. Additionally, quantifying responses to more perturbations across a larger set of cell lines would be essential for constructing more comprehensive models that are applicable to different cell types. Moreover, cell-to-cell variability, which reflects biologically relevant differences in cell cycle dynamics and drug responses, could explicitly be addressed in future work by applying nonlinear mixed-effects models that incorporate both fixed and random effect parameters [61,62], or by using stochastic modeling approaches [63–66] that capture these fluctuations more directly. To this end, the pseudo-time ordering approach we developed could be adapted to stochastic modelling framework and work with data from multiple cell-lines.

The approaches we developed for integration of perturbation data and training of mathematical models are general and can be combined with other pseudo time-ordering approaches that involve combinations of embedding and ordering steps. At the same time, we anticipate that the choice of pseudo-time approach may substantially impact the downstream training of mathematical models. For example, pseudo-time embedding of a dataset featuring more perturbation conditions suggested a much more complicated molecular architecture of the cell cycle [67], but this dataset has yet to be integrated with a mathematical model. As we decided that a thorough exploration of the relationship between the particular pseudo-time embedding and the resulting mathematical model is beyond the scope of this study, we closely followed the data preprocessing and pseudo-time ordering strategy proposed in the original study that generated the investigated dataset, as there was insufficient evidence to challenge the assumptions of the employed methods or the quality of the reconstructed trajectories – a common issue with any pseudo-time approach. This issue could be addressed by evaluating the prediction accuracy of mathematical models trained on reconstructed trajectories using time-series data. Our demonstration of practical and structural identifiability for all but two model parameters, along with accurate predictions under growth factor starvation conditions, builds confidence that model training sufficiently constrains predictions, which is essential to make respective evaluations of prediction accuracy informative.

While the proposed framework proved robust when applied to MCF-10A data, we observed some limitations in capturing cyclin A and B dynamics with high precision. Specifically, model simulations for these markers showed minor deviations from the experimental data for a few data points, likely due to either unmodeled regulatory mechanisms or the limitations of the CMD-based trajectory reconstruction itself. This limitation becomes more pronounced in the case of biopsy data, where the CMD ordering method, though effective in providing ordered trajectories in ER+ cells, remains an approximation and cannot be considered a fully precise representation of true cell cycle progression. The increased complexity, noise, and heterogeneity in biopsy samples (compared to cell line data) can compromise the accuracy of pseudo-time ordering and downstream modeling. These factors should be considered when interpreting model outputs from such datasets.

Despite its limitations, our data-integration strategy represents an initial step toward unifying cell cycle insights across diverse cell types and states. Mass cytometry [68] or the Dye Drop imaging method [69], which can measure single cell states using similar markers across hundreds of conditions, could further illuminate the cell cycle landscape and help refine drug dosing and administration for cell cycle arrest. Ultimately, this approach could pave the way for optimization of drug combinations and timing in therapeutic settings.

## Materials and methods

### Data processing

Data from MCF-10A cell line (epithelial cell line from the mammary gland) untreated and treated with palbociclib and nocodazole from the work by Gaglia et al. [9] were used. Even though Gaglia et al. proposed a cell cycle vector composed of cell cycle markers measured in the study (cyclin D, cyclin E, cyclin A, cyclin B, pRB, CDT1, geminin (Gem), p21, p27, Ki67, PCNA and MCM2 proteins), the CMD scaling was performed considering only a subset of available markers.

In this work we analysed the impact of marker selection on the moving median absolute deviation (movMMAD) of the constructed dynamics obtained by performing CMD. Using all cell cycle markers and removing individual markers we observed and selected the set of parameters with minimum movMMAD. The movMMAD was computed using the *movmad* function of MATLAB [70]. To assess significative differences, statistical tests were performed using the Benjamini & Hochberg (1995) procedure for controlling the false discovery rate for multiple-testing using the *ttest* and *fdr_bh* functions of MATLAB [43]. Subsetting of markers was only applied when performing CMD scaling, not when computing moving absolute deviations. Then, to make the CMD scaling transformations of treated cells comparable with those of untreated cells, a joint CMD embedding of all untreated and small subsets (5%) of treated data was performed repeatedly. The computational analysis was performed using the MATLAB 2023b software [70]. Functions to generate CMD scaling and fit data to a circle from the work of Gaglia et al. [9] were also employed.

To enable comparison between different time inference approaches, we employed the equation proposed by Kafri et al. [71] (Equation 1) to estimate the time vector (t) associated with the marker trajectories:

$$t = \frac{T}{\log(2)} \log\left(\frac{2N}{2N-r}\right)$$

(1)

Where T is the doubling time, N is the total number of cells and r is the number of cells younger than t. For this approach, we used the moving median to the inferred time vector that corresponded to the interpolations of the moving average of marker concentrations of cell cycle markers.

Additionally, CMD scaling method was applied to biopsy data from ER+ (Case 08 and Case 06, N = 5605 untreated/treated cells), HER2+ (N = 6620 untreated and 4695 post-treated cells) and TNBC cells (12000 sampled cells for pre- and on-treatment phases and 2250 cells for the post-treatment period). CMD embedding was performed for each data set for on- and post-treatment data when available. The mode of the angular distribution on- and/or post-treatment was obtained and plotted alongside cell cycle marker dynamics using the original ergodic approach (Cyclins D, A, B, and E when available, or Geminin when Cyclin E was unavailable).

### Cell cycle modeling

The development of the model was based on the complex and skeleton models developed by Gérard & Goldbeter [33,37] by incorporating the main regulations of the synthesis and degradation of CDK-cyclins, CDK inhibitors p21/p27 activity (modelled as one entity), the E2F regulatory mechanisms, and palbociclib and nocodazole effect.

Notably, unlike the skeleton model of Gérard & Goldbeter, the proposed model incorporates the inhibitory activity of p21/p27 on CDK-cyclins' activity and does not include the Cdc20 protein activity. Overall, the model consists of six differential equations that describe the dynamics of $p21$, cyclin D ($Cd$), the active form of E2F ($E2F_a$), cyclin E ($Ce$), cyclin A ($Ca$) and cyclin B ($Cb$). Furthermore, in contrast to the skeleton and complex models of Gérard & Goldbeter, this model incorporates the effect of CDK/cyclin B on the degradation of p21, as well as its indirect inhibitory effect on $E2F_a$ activation, allowing a more comprehensive understanding of the effect of $Cb$ on the entire system.

The relationships between the proteins were described by variable multiplication or, when a relevant protein-mediated reaction is involved, by the multiplication of Michaelis Menten effect equations for both direct and indirect inhibition and activation effects. Thus, the model consists of the following ordinary differential equations:

$$\frac{dp21}{dt} = V_{sp} \cdot E2F_a - V_{sp} \cdot p21 \cdot \left( \frac{C_b}{k_x + C_b} \right) \cdot \left( \frac{1}{1 + \mathbf{\mathit{IE}}_n * \boldsymbol{\delta_{noco}}} \right) \tag{2}$$

$$\frac{dC_d}{dt} = V_{sd} \cdot GF \cdot E2F_a - V_{sd} \cdot \left( \frac{C_d}{C_d + k_{dd}} \right) \tag{3}$$

$$\frac{dE2F_a}{dt} = (E2F_t - E2F_a) \cdot \left( \frac{k_{piii}}{k_{piii} + C_b} \right) \cdot \left( \frac{C_d}{C_d + k_{dd1}} \cdot \boldsymbol{\delta_{palbo}} + C_e \cdot \frac{k_{pi}}{k_{pi} + p21} \right) - E2F_a \cdot C_a \cdot \left( \frac{k_{pii}}{k_{pii} + p21} \right) \tag{4}$$

$$\frac{dC_e}{dt} = V_{se} \cdot E2F_a - V_{se} \cdot \left( \frac{C_e}{C_e + k_{de}} \right) \cdot C_a \cdot \left( \frac{k_{pii}}{k_{pii} + p21} \right) \tag{5}$$

$$\frac{dC_a}{dt} = V_{sa} \cdot E2F_a - V_{sa} \cdot \left( \frac{C_a}{C_a + k_{da}} \right) \tag{6}$$

$$\frac{dC_b}{dt} = V_{sb} \cdot C_a \cdot \left( \frac{k_{pii}}{k_{pii} + p21} \right) - V_{sb} \cdot \left( \frac{C_b}{C_b + k_{db}} \right) \tag{7}$$

The first equation of the model describes the p21 dynamics, Equations 3, 5, 6, and 7 describe the cyclin dynamics, and Equation 4 the active form of E2F ($E2F_a$) dynamics. As in the complex model of Gérard & Goldbeter, p27 and p21 were modelled as one entity (CDK inhibitors), but all detailed regulations of p21, cyclins, E2F, and pRBp were not fully incorporated. In this proposed model, the $p21$ equation includes the upregulation of $p21$ expression by $E2F_a$ [72], and the $p21$ degradation process affected by the action of CDK1/cyclin B since the cyclin B/Cdk1 complex contributes to the formation of active APC/C(Cdc20), which promotes the degradation of $p21$ [54,55]. Then, cyclin equations were described as Gérard and Goldbert skeleton model [37], but the inhibitory effects of $p21$ on cyclins E, A, and B (cyclin/CDK) activity were also included [73]. Moreover, a growth factor parameter (GF) instead of a Michaelis Menten equation was included in the $Cd$ equation to avoid parameter non-identifiability (GF = 1 was fixed for cell cycle proliferation dynamics due to the lack of data at different levels of growth factors).

For E2F, we assume its total amount ($E2F_t$) remains constant, reflecting a steady-state condition where production and degradation are in equilibrium. However, E2F oscillates between active ($E2F_a$) and inactive states ($E2F_i$), allowing to incorporate the E2F regulation using only one differential equation for its active form. The production rate of $E2F_a$ depends on the phosphorylated state of E2F and pRB-complexed inactive states of the E2F protein ($E2F_i = E2F_t - E2F_a$). E2F phosphorylation is promoted by CDK/cyclin A, therefore it was included on the right side of the $E2F_a$ equation. Positive regulations of cyclin D and E activity (CDK/cyclin) on the positive side of $E2F_a$ equation were included due to both CDK/cyclin complexes phosphorylate pRB, which leads to an increase in $E2F_a$ levels (phosphorylated pRB decreases its affinity for E2F, leaving it free in its active form) [33]. Moreover, the indirect inhibitory effect of CDK/cyclin B on pRB dephosphorylation is included on the right side of the $E2F_a$ equation. Since CDK/cyclin

B mediates the inactivation of the protein phosphatase PP2A, which dephosphorylates pRB, it inhibits the formation of $E2F_i$ ($E2F_t - E2F_a$) [52,53].

Finally, inhibitory effect expressions (in bold) were incorporated only under cell cycle arrest conditions into the $p21$ equation for nocodazole and into the $E2F_a$ equation for palbociclib. Presence of nocodazole was incorporated by setting $\delta_{noco} = 1$, and presence of palbociclib with $\delta_{palbo} = 0$. Nocodazole increases the levels of $p21$, which promotes the G1 arrest at high doses [51], and palbociclib is a CDK4/6 inhibitor, thus inhibiting the action of CDK4/6-cyclin D on pRB, increasing the $E2F_a$ levels [56].

### Parameter estimation, confidence intervals and simulations

The parameter estimation was performed using the Data2Dynamics modelling environment (2024 version) in MATLAB 2023b [74,75]. The model was calibrated simultaneously to all the experimental conditions: untreated (marker dynamics) and treated conditions (cell cycle arrest states). Four replicates of the same data were used to ensure that the model simulations recapitulate the oscillations and that the perturbed data reaches a steady state ~24 hours after drug treatment (corresponding to approximately one cell cycle of the MCF-10A cell line) [47]. In order to constrain the space of feasible parameter values, we used two initial conditions to fit the model for both untreated and treated datasets. Initial conditions were set according to the mode of the main subpopulations of cell cycle angles (**Fig 1C**). Due to narrow oscillations of the variables, cell arrest states were not incorporated as steady state constraints but as points of equal value (corresponding to the arrest state) after one cell cycle of drug treatment.

Regarding the training data, marker dynamics corresponding to 12,000 MCF-10A cells were obtained after data pre-processing, but only ten points of the moving median dynamics per cell cycle were used to fit the model (proliferative dynamics). Parameters GF and $E2F_t$ were set to 1 to avoid model fitting problems due to parameter non-identifiability, and offset parameters were added to experimental data ($offsetp21$, $offsetCd$, $offsetCe$, $offsetCa$, and $offsetCb$) to account for background intensities. Initial values of E2F were added as parameters for estimation ($initE2FG1S$, $initE2FG2M$). Profile likelihoods were generated by applying the profile likelihood method (PL algorithm) implemented in Data2Dynamics [58,74]. Moreover, a structural identifiability analysis was performed using the *StrucID* tool from the Data2Dynamic environment, which is based on sensitivity matrix analysis [57]. All simulations and figures were performed in MATLAB 2023b using the *ode15s* integration solver [70]. Finally, the predicted cellular arrest state for the starvation condition was estimated using the *dsearchn* function of MATLAB [70], which returns the index of the closest data vector for a given vector of values using Euclidean distance. In this case, the data of the dynamics in the cell proliferation state and the vector of values predicted by the model when GF = 0.6 after four cell cycles were compared to define the predicted cell arrest phase. MATLAB scripts for model calibration, simulation, and data analysis can be reached at https://www.synapse.org/#!Synapse:syn55267562/files/.

### Supporting information

**S1 Fig. Moving absolute deviations of the normalized RFU of marker dynamics obtained after performing CMD scaling with all markers (All) and all markers except one of them (-cycD, -pRB, -cycE, -cycA, -cycB, -CDT1, -Gem, -p21, -p27, -Ki671, -PCNA and -MCM2).**
(TIF)

**S2 Fig. Moving absolute deviations of the normalized RFU of marker dynamics obtained after performing CMD scaling with all available markers except one of them (-cycD, -pRB, -cycE [only available for TNBC sample], -cycA, -cycB, -CDT1, -Gem, -p21, -p27, -Ki671, -PCNA and -MCM2) from biopsy samples.** (A) ER+ cells. (B) TNBC cells. (C) HER2 + cells.
(TIF)

**S3 Fig. Fitted model results integrating untreated and treated data sets using the equation proposed by Kafri et al. to infer cell dynamics.** (A) Model fitting to cell cycle marker dynamics obtained from untreated and treated MCF-10A cells processed data using a non-ergodic approach to infer the dynamics. (B) Profile likelihood (PL) for each estimated parameter (black dotted line) and estimated parameter value (gray asterisk).
(TIF)

**S4 Fig. Data processing results using biopsy data.** (A) CMD results for ER+ cells pre- (grey) on- (red) and post-treatment (blue). (B) Polar plot of normalized Cyclin D, A, B and Geminin dynamics in untreated ER+ cells, with angles mode during tamoxifen/aromatase inhibitor treatment (red line). (C) CMD results for TNBC cells pre- (grey) on- (red) and post-treatment (blue). (D) Polar plot of normalized Cyclin D, E, A and B dynamics in untreated TNBC cells, with angles modes of cells on-treatment with paclitaxel (red) and post-treatment with doxorubicin and cyclophosphamide (blue). (E) CMD results for HER2+cells pre- (grey) and post-treatment (blue). (F) Polar plot of normalized Cyclin D, A, B and Geminin dynamics in untreated HER2+cells, with angles mode post-treatment with pertuzumab and ado-trastuzumab emtansine (blue line).
(TIF)

**S5 Fig. Data processing results using the treated data sets to perform CMD at once (on the left) versus using both untreated and treated data sets to perform CMD at once (on the right).** (A, D) CMD transformation results for MCF-10A cells treated with nocodazole on the left and palbociclib on the right. Every dot corresponds to a single cell, in green nocodazole treated cells, in purple palbociclib treated cells and in grey untreated cells. (B, E) Histogram of CMD angles distribution for nocodazole on the left and palbociclib on the right show unimodal distributions that were used to map the corresponding cell cycle arrest times. (C, F) Polar coordinate plot of normalized cyclin dynamics (untreated cells) and cell arrest times for nocodazole and palbociclib. Radial coordinate corresponds to cell cycle time from untreated cell dynamics. Polar coordinate corresponds to moving median of marker levels min-max normalized to the interval [0,100]. Cell cycle phases between checkpoints were manually annotated based on cyclin dynamics obtained previously.
(TIF)

**S6 Fig. Angle distribution of untreated cells.** Angle distribution results using data processing results from CMD performed with untreated data only (gray) versus untreated+treated datasets used at once. Light green for untreated+nocodazole datasets (A), and light purple for untreated+palbociclib datasets (B).
(TIF)

**S7 Fig. Log-likelihood profiles of the fitted model using untreated and treated data sets.** Profile likelihood (PL) for each estimated parameter (black line) and estimated parameter value (gray asterisk).
(TIF)

**S8 Fig. Log-likelihood profiles of the fitted model using only the untreated data set.** Profile likelihood (PL) for each estimated parameter (black line) and estimated parameter value (gray asterisk).
(TIF)

**S9 Fig. Reduction of cell cycle duration by reducing GF value.** Predicted dynamics across six cell cycles of the state variables (A) $p21$, (B) $Cd$, (C) $E2F_a$, (D), $Ce$ (E) $Ca$ and (F) $Cb$ sing GF = 100% (black lines) and GF = 95% (dashed gray lines).
(TIF)

**S10 Fig. Cell cycle arrest prediction for low growth factor values using the model fitted with the untreated data set only.** (A) Predicted dynamic of the state variables $p21$, $Cd$, $E2F_a$, $Ce$, $Ca$ and $Cb$ for different starting points of the cell cycle (initial conditions) lead to the same cell arrest state for GF = 0.0001%. (B) Polar coordinate plot of normalized

cyclin dynamics (untreated cells) and arrest time predicted for GF = 0.0001% at S phase. Polar coordinate corresponds to moving median of marker levels min-max normalized to the interval [0,100]. Cell cycle phases between checkpoints were manually annotated based on cyclin dynamics obtained previously. (C) Simulations for *Ce*, *Cd* and *Cb* variables at different growth factor (GF) levels (0.0001, 0.001, 1 and 100%) reveal a limit cycle for high GF values (≥ 1%) and a cell arrest at lower GF values.
(TIF)

**S1 Table. Estimated parameters of the cell cycle model.**
(XLSX)

**S1 Text. Processing of multiplexed immunofluorescence data from biopsy samples.**
(DOCX)

## Acknowledgments

We thank Giorgio Gaglia for his valuable comments, which helped us better understand and improve the data processing method.

## Author contributions

**Conceptualization:** Javiera Cortés-Ríos, Peter Karl Sorger, Fabian Fröhlich.

**Formal analysis:** Javiera Cortés-Ríos, Fabian Fröhlich.

**Funding acquisition:** Maria Rodriguez-Fernandez, Peter Karl Sorger.

**Investigation:** Javiera Cortés-Ríos.

**Methodology:** Javiera Cortés-Ríos, Fabian Fröhlich.

**Project administration:** Maria Rodriguez-Fernandez, Peter Karl Sorger.

**Software:** Javiera Cortés-Ríos, Fabian Fröhlich.

**Supervision:** Fabian Fröhlich.

**Writing – original draft:** Javiera Cortés-Ríos, Fabian Fröhlich.

**Writing – review & editing:** Javiera Cortés-Ríos, Maria Rodriguez-Fernandez, Peter Karl Sorger, Fabian Fröhlich.

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
