## [Decision Letter · Decision Letter 0]

24 Apr 2025

PCOMPBIOL-D-25-00318

Dynamic Modeling of Cell Cycle Arrest Through Integrated Single-Cell and Mathematical Modelling Approaches

PLOS Computational Biology

Dear Dr. Fröhlich,

Thank you for submitting your manuscript to PLOS Computational Biology. After careful consideration, we feel that it has merit but does not fully meet PLOS Computational Biology's publication criteria as it currently stands. Therefore, we invite you to submit a revised version of the manuscript that addresses the points raised during the review process.

Please submit your revised manuscript within 60 days Jun 24 2025 11:59PM. If you will need more time than this to complete your revisions, please reply to this message or contact the journal office at ploscompbiol@plos.org. Please include the following items when submitting your revised manuscript:

We look forward to receiving your revised manuscript.

Kind regards,

Ovidiu Radulescu, PhD

Academic Editor

PLOS Computational Biology

Pedro Mendes

Section Editor

PLOS Computational Biology

**Journal Requirements:**

4) Please amend your detailed Financial Disclosure statement. This is published with the article. It must therefore be completed in full sentences and contain the exact wording you wish to be published.

5) Please ensure that the funders and grant numbers match between the Financial Disclosure field and the Funding Information tab in your submission form. Note that the funders must be provided in the same order in both places as well. Currently " Ludwig Cancer Center at Harvard" is missing from the Funding Information tab.

**Reviewers' comments:**

Reviewer's Responses to Questions

Reviewer #1: The paper by Cortés-Ríos et al presents an integrated framework combining single‐cell multiplexed immunofluorescence data with mathematical modelling to decipher cell cycle dynamics under both oscillatory and arrest conditions. The authors extend the cc‐CMD pseudo‐time ordering method to integrate data from multiple experimental conditions, including treatments with palbociclib and nocodazole. By incorporating regulatory interactions such as p21/p27 inhibition of CDK‐cyclins and the impact of cyclin B on p21 degradation, the model successfully recapitulates the cell cycle progression of untreated MCF 10A cells and accurately predicts arrest states corresponding to early G1. Model calibration using both untreated and drug‐treated datasets reduced parameter uncertainty and yielded biologically plausible simulations. The predictive capacity of the model was further validated under growth factor starvation, demonstrating its potential for extrapolating cell cycle arrest conditions not included in the training set. The paper presently a sensible approach though I am not 100% sure of some of the assumptions made and the extensibility of the approach to other datasets.

1. The method is inherently based on the assumption of ergodicity, i.e. that the cell population is representative across all phases in proportion to the time spent in each phase—which may not hold true for unsynchronized cells or those grown under non‐ideal conditions. If this assumption fails, the inferred cell cycle times and phase transitions could be significantly skewed. Can the authors test the effect on their method’s output by using simulated data instead of experimental data, e.g. using the stochastic simulation algorithm, that is not in quasi-steady state conditions and that hence does not obey the ergodic hypothesis.

2. Another potential issue is the reliance on a single MCF-10A cell line and a restricted marker panel, which raises concerns about the generalizability of the findings to other cell types or biological contexts. Would be useful to apply to data beyond this particular cell line to remove these concerns.

3. The method appears to completely ignore the inherent large cell-to-cell variability in single cell data (as they use the moving median for the protein levels). While noise can be seen as a nuisance, actually there are many studies (especially in the scRNA sequencing literature) which point out that it possesses biological information that is better used rather than discarded. The authors should discuss how they would include this variability in the protein levels and the anticipated effect on their method and on the main results reported in this paper. In this context, the literature on the use of stochastic models of the cell-cycle (especially those used in conjuction with single-cell data) would probably be useful to consult and to reference appropriately.

Reviewer #2: The title of the manuscript is misleading: it should highlight that the main contribution of this work is methodological, and not on modeling of cell cycle arrest. As far as I understand, the authors show that using pseudotime techniques on multiple experimental data via appropriate training provides accurate and relevant results, when applied to the modeling of cell cycle arrest. This is certainly an important result, that would deserve publication, provided that its relevance is better motivated.

My detailed comments:

Lines 101-102, the authors claim to "develop novel model training techniques". Reading through the manuscript, it is unclear what is new, since the cc-CMD method is not new and already proved to be useful. Moreover, it is not clear what the extension of this method consists in. Since this is, in my opinion, the core of the current work, it should be clearer and discussed in more depth (advantages, limitations, etc.).

Lines 160-162, the authors claim their results "confirm" the relevance of CMD-based pseudo-time techniques, yet, as mentioned above, this was already known. Can the authors clarify what is confirmed by their results?

Lines 231-232, the authors mention that "the simulations for cyclin B and cyclin A were slightly outside the median absolute deviation range for a few datapoints", could they justify this result or at least hypothesize why this is observed? It is probably the most interesting unexpected result of their work, I believe it deserves more attention.

Lines 271-272, the authors wrote "our results demonstrate that training on data from multiple experimental conditions make it possible to reduce parameter uncertainty", I must say that it seems obvious: if you train on more data you reduce the uncertainty. What exactly is noticeable in the results? Is it that it actually works? Or how better it works, comapred to previous methods? Please clarify.

In the discussion, the authors argue (lines 304-306) that they "decided that a thorough exploration of the relationship between the particular pseudo-time embedding and the resulting mathematical model is beyond the scope of this study", so they do not investigate further the impact of using the pseudo-time technique in the resulting training of the models. I do believe, on the contrary, that this is in the scope of the manuscript. One cannot claim that a method is better than previous ones without investigating its limitations.

Finally, it is overall unclear what are the contributions of the authors: it seems it is a training accounting for multiple (here 2) conditions as well as a modificaton of the cell cycle model. Yet this is unclear, and I suggest it is clarified before a final decision is taken.

**Have the authors made all data and (if applicable) computational code underlying the findings in their manuscript fully available?**

Reviewer #1: Yes

Reviewer #2: Yes

PLOS authors have the option to publish the peer review history of their article (what does this mean? ). If published, this will include your full peer review and any attached files.

**Do you want your identity to be public for this peer review?** For information about this choice, including consent withdrawal, please see our Privacy Policy .

Reviewer #1: No

Reviewer #2: No

**Figure resubmission:**
---

## [Decision Letter · Decision Letter 1]

24 Sep 2025

Dear Mr Fröhlich,

We are pleased to inform you that your manuscript 'Dynamic Modelling of Cell Cycle Arrest Through Integrated Single-Cell and Mathematical Modelling Approaches' has been provisionally accepted for publication in PLOS Computational Biology.

Best regards,

Ovidiu Radulescu, PhD

Academic Editor

PLOS Computational Biology

Pedro Mendes

Section Editor

PLOS Computational Biology

Reviewer's Responses to Questions

**Comments to the Authors:**

Reviewer #1: The authors have done substantial changes to the manuscript in response to all reviewer comments. I am happy to suggest the paper's acceptance in its present form.

Reviewer #2: The authors have correctly revised their manuscript, the revised version is much clearer. I am satisfied with the authors answers to my comments. However, I believe the paper would have benefited from a more detailed discussion of the difficulties and challenges posed by the initial question and answered by the authors' work, and how their contribution would be relevant for a large community.

**Have the authors made all data and (if applicable) computational code underlying the findings in their manuscript fully available?**

Reviewer #1: None

Reviewer #2: Yes

PLOS authors have the option to publish the peer review history of their article (what does this mean? ). If published, this will include your full peer review and any attached files.

**Do you want your identity to be public for this peer review?** For information about this choice, including consent withdrawal, please see our Privacy Policy .

Reviewer #1: No

Reviewer #2: No

---

## [Editor Report · Acceptance letter]

PCOMPBIOL-D-25-00318R1

Dynamic Modelling of Cell Cycle Arrest Through Integrated Single-Cell and Mathematical Modelling Approaches

Dear Dr Fröhlich,

I am pleased to inform you that your manuscript has been formally accepted for publication in PLOS Computational Biology. Your manuscript is now with our production department and you will be notified of the publication date in due course.

With kind regards,

Anita Estes
